# Free Fatty Acid Receptors (FFARs) in Adipose: Physiological Role and Therapeutic Outlook

**DOI:** 10.3390/cells11040750

**Published:** 2022-02-21

**Authors:** Saeed Al Mahri, Shuja Shafi Malik, Maria Al Ibrahim, Esraa Haji, Ghida Dairi, Sameer Mohammad

**Affiliations:** 1Experimental Medicine, King Abdullah International Medical Research Center (KAIMRC), King Saud Bin Abdulaziz University for Health Sciences (KSAU-HS), Ministry of National Guard Health Affairs (NGHA), Riyadh 11426, Saudi Arabia; almahrisa@ngha.med.sa (S.A.M.); maliksh@ngha.med.sa (S.S.M.); alibrahimmaria1@gmail.com (M.A.I.); hajiesraam@gmail.com (E.H.); gsdairi@hotmail.com (G.D.); 2Physiology Department, College of Medicine, King Saud University, Riyadh 11362, Saudi Arabia; 3Deanship of Scientific Research, Umm Al-Qura University, Makkah 21961, Saudi Arabia

**Keywords:** adipose tissue, G-protein-coupled receptors, free fatty acid receptors, thermogenesis, adipogenesis

## Abstract

Fatty acids (FFAs) are important biological molecules that serve as a major energy source and are key components of biological membranes. In addition, FFAs play important roles in metabolic regulation and contribute to the development and progression of metabolic disorders like diabetes. Recent studies have shown that FFAs can act as important ligands of G-protein-coupled receptors (GPCRs) on the surface of cells and impact key physiological processes. Free fatty acid-activated receptors include FFAR1 (GPR40), FFAR2 (GPR43), FFAR3 (GPR41), and FFAR4 (GPR120). FFAR2 and FFAR3 are activated by short-chain fatty acids like acetate, propionate, and butyrate, whereas FFAR1 and FFAR4 are activated by medium- and long-chain fatty acids like palmitate, oleate, linoleate, and others. FFARs have attracted considerable attention over the last few years and have become attractive pharmacological targets in the treatment of type 2 diabetes and metabolic syndrome. Several lines of evidence point to their importance in the regulation of whole-body metabolic homeostasis including adipose metabolism. Here, we summarize our current understanding of the physiological functions of FFAR isoforms in adipose biology and explore the prospect of FFAR-based therapies to treat patients with obesity and Type 2 diabetes.

## 1. Introduction

Free fatty acids (FFAs) play a vital role as energy substrates and form key components of cellular membranes. Diet is the main source of long- and medium-chain free fatty acids whereas short chain fatty acids are produced as a result of the bacterial fermentation process in the gut. Over the past few years, FFAs have emerged as important signaling molecules in the regulation of metabolic homeostasis. FFAs have been shown to activate cell surface receptors known as Free Fatty Acid Receptors (FFARs). FFARs belong to the family of G-Protein-Coupled Receptors (GPCRs) that signal via the activation of hetero-trimetric G-Protein complex. GPCRs are characterized by seven signature transmembrane domains, an extracellular N-terminus, and an intracellular C-terminus [1,2]. Around 800 GPCRs have been identified in humans, of which more than half have sensory functions, including olfactory (391), Vision (10), taste (33), and pheromone receptors [1,3,4]. The remaining 356 non-sensory GPCRs mediate the signaling of a variety of ligands ranging from small molecules and metabolites to peptides and large proteins [5]. Free fatty acids have also been shown to activate signaling cascades involving GPCRs. To date, four FFARs have been identified that act as ligands for FFAs based on their chain length (Figure 1). FFAR1 (GPR40) and FFAR4 (GPR120) are activated by long-chain fatty acids like palmitate, oleate, and linoleate, whereas FFAR2 (GPR43) and FFAR3 (GPR41) are mainly activated by short-chain fatty acids like acetate, butyrate, and propionate [6,7,8,9]. FFARs are widely expressed throughout the human body and have been shown to regulate multiple biological processes. FFAR-mediated signaling has been implicated in metabolic processes like insulin secretion from pancreatic beta cells, incretin secretion from entero-endocrine cells, regulation of food intake, adipose tissue biology, and many more [6,7,10,11,12,13]. These receptors are considered attractive therapeutic targets for metabolic disorders like obesity and type 2 diabetes. Several agonists of FFARs have been developed and tested in animal models, as well as in human trials [14,15,16,17]. Here we focus on the role of FFARs in regulating various aspects of adipose biology, including adipogenesis, lipid, and glucose metabolism, and explore the prospect of FFAR-based therapies to treat patients with metabolic disorders.

## 2. FFAR2 and FFAR4 Are Highly Expressed in Adipose Tissue

FFAR isoforms are expressed throughout the body and regulate many biological processes [18,19,20,21,22,23,24,25,26,27,28,29,30,31,32]. Two FFAR isoforms have substantial levels of expression in adipose tissue and adipocytes: FFAR2 (also known as GPR43) and FFAR4 (also known asGPR120). Several independent studies have documented the expression of FFAR2 and FFAR4 in human and mouse adipose tissue and cultured adipocytes [8,18,19]. FFAR3 (GPR41) is also expressed in human adipose tissue but to a lesser extent compared to FFAR2. On the other hand, FFAR3 mRNA or protein has not been detected in mouse adipose tissue and cultured adipocytes [20,21,22]. FFAR1 has not been detected in human or mouse adipose tissue [23]. Table 1 lists the major sites of FFAR expression in the human body.

## 3. Role of FFAR2 in Adipose Metabolism and Energy Homeostasis

Numerous studies have demonstrated that FFAR2 is expressed in human and mouse white adipose tissue (WAT), and also in the murine adipocyte cell line 3T3L1 [33,34,35]. Hong et al. reported that FFAR2 was highly expressed in adipocytes, with a much lower expression in stromal-vascular cells [34]. In addition, FFAR2 expression was upregulated during the adipogenic differentiation of 3T3L1 cells. The authors also observed that FFAR2 expression was elevated in adipose tissue of high-fat diet mice prompting them to explore the role of FFAR2 in the adipogenic process. The authors demonstrated that natural agonists of FFAR2, acetate and propionate promote the adipogenesis process and this effect was mediated by FFAR2. On the contrary, Dewulf et al. reasoned that FFAR2 was not involved in human adipogenesis based on their observation that FFAR2 expression was not upregulated in white adipose of obese individuals and the inability of FFAR2 agonists to induce differentiation of human pre-adipocytes into mature adipocytes [36]. This assessment was strengthened by another study, which showed that propionate or acetate has no effect on adipogenesis in 3T3-L1 cells [37]. The authors demonstrated that FFAR2 is not expressed until approximately 48 h into the differentiation process of 3T3 L1 adipocytes, and supplementing differentiation cocktail with acetate or propionate did not have any effect on the differentiation process. Yet another study by Ivan et al. showed that FFAR2 had an inhibitory role in adipogenic differentiation of human Mesenchymal Stem cells (MSCs) [38]. The authors reported that propionate and a synthetic agonist of FFAR2 suppressed adipogenic differentiation and that this effect was mediated by FFAR2. In total, the role of FFAR2 in adipogenesis is still a matter of debate, and more studies on both human and murine adipocytes are needed to get a clearer picture of the involvement of FFAR2 in adipogenesis.

While the impact of FFAR2 on adipogenesis is still unclear, many studies have shown that FFAR2 activation is associated with inhibition of lipolysis. Using murine adipocyte cell line 3T3 L1, Hong et al. demonstrated that acetate and propionate inhibited isoproterenol-induced lipolysis and this effect was abolished in FFAR2 deficient adipocytes [34]. Another study confirmed the anti-lipolytic activity of FFARs in both 3T3L1 adipocytes as well in primary adipocytes [33]. The authors further showed that in vivo infusion of acetate reduced the circulating free fatty acid levels in mice. The anti-lipolytic effect of acetate was completely absent in FFAR2 knockout mice supporting the role of FFAR2 in this process. Lee et al. also demonstrated that acetate suppressed lipolysis in 3T3 L1 adipocytes in a dose-dependent manner [39]. FFAR2-specific synthetic agonist developed by Wang et al. inhibited lipolysis via the activation of FFAR2 in murine adipocytes [40]. This synthetic agonist reduced circulating FFA levels when administered to mice. Several other synthetic agonists of FFAR2 have been shown to inhibit lipolysis in adipocytes.

The effect of FFAR2 on adipose insulin sensitivity and whole-body metabolic homeostasis has been the subject of several independent studies but the results are inconclusive [41]. Two independent studies point to a positive role played by FFAR2 in the regulation of metabolic homeostasis. Tolhurst et al. showed that *ffar2* knockdown mice fed on a normal diet had impairment in glucose tolerance [42]. Consistent with this study, Kimura et al. reported that *ffar2* knockout mice were obese and glucose-intolerant when fed with a standard chow diet. Additionally, mice overexpressing ffar2 in adipose tissue remained lean and glucose tolerant even when fed a high-fat diet [43]. Conversely, Bjursell et al. studied the effect of FFAR2 in lipid and energy metabolism by using FFAR2 knockout animals. The authors showed that the loss of FFAR2 was not associated with any metabolic abnormality in mice fed a normal diet. Interestingly, FFAR2 knockout mice were protected from HFD-induced obesity and metabolic abnormalities suggesting a negative influence of FFAR2 on metabolic homeostasis [44]. The reasons for the discrepancy in the outcome of studies on FFAR2 knockout mice are unclear but could be due to the different genetic backgrounds of mice. Bjursell et al. [44] used C57BL/6 mice for their study, whereas both Tolhurst et al. [42] and Kimura et al. [43] used 129/SvEv background mice.

Overall, FFAR2 appears to have considerable functional significance in adipose tissue metabolism and whole-body energy homeostasis but more work is needed to address the inconsistencies reported by different research groups.

## 4. Role of FFAR4 (GPR120) in Adipogenesis and Adipose Metabolism

Multiple lines of evidence suggest that FFAR4 plays a critical role in adipose tissue metabolism. First, in humans, the expression of FFAR4 is significantly higher in adipose tissue of obese subjects compared to lean healthy individuals [45]. In addition, a deleterious variant of FFAR4 (R270H) is associated with an increased risk of obesity and increased fasting glucose levels in human subjects of European origin [45,46]. The R270H FFAR4 variant has also been shown to act together with dietary fat intake to modulate the risk of type 2 diabetes [47]. Second, FFAR4 is highly expressed in adipocytes and adipose tissues, and shows a marked increase in expression during the adipogenic differentiation of murine and human pre-adipocytes. Third, mice deficient in FFAR4 are more likely to develop obesity, metabolic dysregulation, and insulin resistance, as well as increased inflammation in adipose tissue [45,47]. In 3T3L1 adipocytes, forced reduction of FFAR4 using siRNA was shown to inhibit differentiation and lipid accumulation. In addition, downregulation of FFAR4 also led to impaired insulin signaling via reduction of glucose transporter (GLUT4) and insulin receptor substrate (IRS) expression [48]. On the other hand, activation of FFAR4 by specific agonists has been found to promote adipocyte differentiation [49]. Moreover, treatment of obese/insulin-resistant mice with a specific agonist of FFAR4 results in improved glucose tolerance, enhanced insulin sensitivity, and reversal of metabolic abnormalities [50]. FFAR4 is also highly expressed in brown adipose tissue (BAT) and is upregulated in mice exposed to cold. The activation of FFAR4 leads to increased thermogenesis via increased expression of BAT-specific markers including uncoupling protein 1 (UCP1) [49,51,52]. In addition, FFAR4 knockout mice have impaired browning of subcutaneous WAT in response to cold exposure [52]. Moreover, FFAR agonists have been shown to increase fatty acid uptake and oxidation, augment mitochondrial respiration, and reduce fat mass in mice [53,54]. A recent study demonstrated that induction of FFAR4 signaling elevated the level of the circulating FGF21, and therefore, enhanced BAT activities and browning of WAT in mice [55]. Furthermore, a study showed that *ffar4*-knockout neonatal mice had reduced activity of neonatal BAT and inhibition of thermogenesis, leading to reduced UCP1 expression, fatty acid oxidative capacity, and circulating levels of fibroblast growth factor 21 (FGF21). Therefore, *ffar4*-knockout mice had cold intolerance after birth and their survival was impacted [52]. Figure 2 summarizes the role of FFAR4 in adipose tissue metabolism in humans, mice, and cultured adipocytes.

## 5. The Downstream Signal Mediated by FFAR2 and FFAR4 in Adipose Tissue

GPCRs signal through the activation of trimetric G-protein complex. Activation of a GPCR by its ligand results in the exchange of GDP with GTP in the G-protein complex, leading to the dissociation of the Gα subunit from the Gβγ subunit. The activated G protein subunits then bind to their respective effectors and activate various signaling pathways (Figure 1). Multiple studies have shown that FFAR2 signals through the Gαi subunit, resulting in the inhibition of the cAMP/PKA pathway [39,40]. FFAR2 has also been shown to signal through Gαq subunit, which leads to an increase of intracellular calcium concentration [Ca^2+^]I and the activation of the MAPK (mitogen-activated protein kinase) cascade [41,42]. Suppression of lipolysis by FFAR2 activation involves Gαi-mediated inhibition of the cAMP/PKA pathway [40]. FFAR2-activated Gαq pathways have been shown to mediate GLP1 secretion from L cells of the intestine, but the significance of this pathway in adipose tissue has not been studied [42]. FFAR4 couples to Gαq, which induces an increase in [Ca^2+^]i levels and activation of MAPK cascade [44,45]. Both pathways (MAPK and [Ca^2+^]i) mediate the FFAR effect on the adipogenesis process [47,48]. FFAR4 does not couple to Gαs or Gαi subunits, and therefore has no effect on the cAMP/PKA pathway in adipocytes. 

## 6. FFAR Agonists in the Treatment of Metabolic Diseases

Ever since the members of the FFAR family were de-orphanized, their role in different biological processes has been extensively studied. In particular, their contribution to metabolic and energy homeostasis has been comprehensively recognized, and they have attracted considerable attention in the drug discovery field [56,57,58,59]. As a result, several ligands of FFARs have been developed to use as therapeutic drugs for metabolic diseases like diabetes and obesity. The list of synthetic ligands of FFAR and their physiological functions are summarized in Table 1. Out of the four isoforms, FFAR1 has emerged as the most targeted receptor of this family. Several ligands targeting FFAR1 have been developed and functionally validated in preclinical in vitro and in vivo studies, and some of them have even reached human clinical trials. The most advanced FFAR1 agonist that has reached human clinical trials is TAK-875, developed by Takeda. TAK-875 had shown promising efficacy and significantly reducing HbA1C levels without causing any hypoglycemic episodes. However, during phase 2 clinical trials, the compound was found to cause liver toxicity, and further development of the drug was discontinued [60,61]. Efforts are being made to develop safer alternatives with good efficacy and with no or little toxicity [16,20,62,63].

Synthetic ligands of FFAR2 have also been developed and evaluated for their therapeutic efficacy in preclinical studies. The first FFAR2-specific agonist, (2S)-2-(4-chlorophenyl)-*N*-(5-fluoro-1,3-thiazol-2-yl)-3-methylbutanamide (CMTB) was discovered by Lee et al. [39]. CMTB was able to inhibit cAMP formation with higher potency than acetate and propionate in CHO cells stably expressing FFAR2. In addition, CMTB inhibited lipolysis in 3T3 L1 adipocytes in a dose-dependent manner through the Gαi pathway. Subsequently, after a detailed investigation of the pharmacology of CMTB, the binding site on FFAR2 was found to be distinct from that of the endogenous ligands. In addition, the signaling responses to CMTB at FFAR2 were not identical to those of endogenous SCFAs [64]. Therefore, there is a need to develop FFAR2 agonists that have biochemical properties similar to the endogenous ligands of FFAR2. Hudson et al. described 3-benzyl-4-(cyclopropyl-(4-(2,5-dichlorophenyl)thiazol-2-yl)amino)-4-oxobutanoic acid (compound1) as a highly potent and selective ortho-ateric ligand of FFAR2. Compound 1 was shown to inhibit lipolysis in both human and murine adipocyte cell lines. In addition, Compound 1 stimulates GLP-1 release from murine STC-1 enteroendocrine cells [65]. Hansen et al. described TUG-1375 as a potent agonist of FFAR2 with favorable pharmacokinetic properties [66]. TUG-1375 was able to inhibit isoproterenol-induced lipolysis in murine adipocytes with a 50-fold more potency as compared to the natural agonist of FFAR2, propionate. None of these ligands have undergone preclinical or clinical trials but these compounds have huge potential to serve as useful tools to study FFA2 function and help in the development of future ligands at FFAR2 with tremendous therapeutic potential [65].

FFAR4 has attracted considerable attention from the pharmaceutical industry due to its multiple functions in metabolic homeostasis. Many FFAR4-specific ligands have been developed to study FFAR4 function and potentially as therapeutic drugs in metabolic diseases. The first reported and extensively studied FFAR4-specific synthetic agonist was TUG-891 [3-(4-((4-fluoro-4′-methyl-[1,1′-biphenyl]-2-yl)methoxy)phenyl)propanoic acid] [67]. The signaling properties of TUG-891 were comparable to natural agonists of FFAR4, and stimulation of human cells with TUG-891 induced Ca^2+^ mobilization, recruitment of β –arrestins, and activation of ERK pathway [67,68]. TUG-891 was also able to induce phosphorylation of FFAR4 followed by receptor internalization. TUG-891 was able to mimic almost all the beneficial properties of natural agonists of FFAR4 including GLP-1 secretion from enteroendocrine cells, augmenting glucose uptake in murine adipocytes, and inhibiting inflammatory mediators from macrophages [66,69]. TUG-891 was also shown to reduce food intake, enhance insulin signaling and reverse insulin resistance in mice. These results indicate great promise for TUG-891 as a potential therapeutic molecule for type 2 diabetes and obesity. Several other FFAR4-specific agonists have been developed and shown to have promising anti-diabetic properties [53,54,70]. The list of FFAR agonists that have been developed for potential therapeutic value is summarized in Table 2.

## 7. Conclusions

Free fatty acid receptor isoforms (FFAR2 and FFAR4) are highly expressed in adipose tissue and have been shown to regulate multiple aspects of adipose metabolism. FFARs have received considerable interest in the last few years and several research groups have provided strong evidence indicating the possibility of using FFARs as novel therapeutic targets to correct metabolic abnormalities associated with obesity and type 2 diabetes. Several synthetic FFAR agonists have been developed and shown to have anti-diabetic effects in preclinical studies. In the coming years, many FFAR agonists are expected to be tested in human clinical trials. However, several key questions regarding the role of FFARs in adipogenesis and adipose metabolism must be answered before they can progress through the trial of drug development. The development of adipose-specific *ffar2* and *ffar4* knockout mice represents an exciting research opportunity and can answer some of these important questions and help in the drug development process.

## Figures and Tables

**Figure 1 cells-11-00750-f001:**
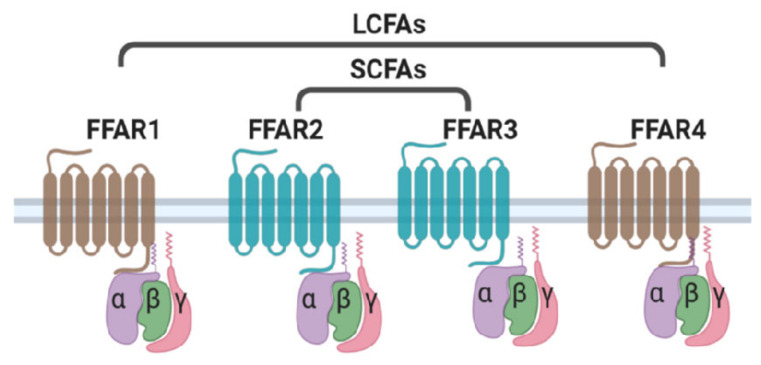
Ligand specificity of free fatty acid receptors. FFAR1 and FFAR4 act as receptors for long chain fatty acids (LCFAs), whereas short chain fatty acids (SCFAs) selectively activate FFAR2 and FFAR3.

**Figure 2 cells-11-00750-f002:**
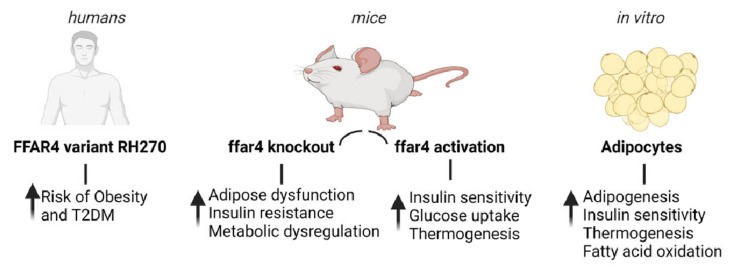
Physiological significance of FFAR4 in adipose tissue metabolism. Evidence from human, mouse, and in vitro studies.

**Table 1 cells-11-00750-t001:** List of FFAR isoforms and their tissue distribution.

Name	Major Expression Sites	References
FFAR1	Pancreatic β-cells, enteroendocrine cells, nerves, immune cells	[7,11,12,13,14]
FFAR2	Adipose, enteroendocrine cells, pancreatic b-cells, immune cells	[25,26,27,28]
FFAR3	Pancreatic β-cells, enteroendocrine cells, nerves	[28,29]
FFAR4	Adipose, enteroendocrine cells, liver, bone, lung, immune cells	[15,16,17]

**Table 2 cells-11-00750-t002:** List of synthetic agonists of FFAR isoforms and their physiological actions.

Agonist Name	Target	Physiological Functions	Reference
TAK-875	FFAR1	Stimulates glucose-dependent Insulin secretion and Improves glycemic control in T2DM patients	[2]
AMG837	FFAR1	Increases insulin secretion and lowers blood glucose levels in mice	[3]
GW-9508	FFAR1	Enhances insulin sensitivity and regulates glucose homeostasis	[4]
TUG-424	FFAR1	Improves glucose tolerance in mice	[5]
AM-1638	FFAR1	Increases insulin secretion and lowers blood glucose levels in mice	[6]
AM-5262	FFAR1	Enhances glucose-stimulated insulin secretion (mouse and human islets) and improves glucose homeostasis in mice	[7,8]
LY2881835	FFAR1	Stimulates insulin secretion from pancreatic β-cells	[9]
MK-2305	FFAR1	Increases glucose-stimulated insulin secretion, resulting in improvement of glucose homeostasis in the diabetic mice	[10]
CMTB	FFAR2	Inhibits lipolysis in murine adipocytes	[11]
TUG-1375	FFAR2	Induces migration of human neutrophils and inhibits lipolysis in murine adipocytes	[12]
Compound 1	FFAR2	Inhibits lipolysis in murine adipocytes	[13]
TUG-891	FFAR4	Stimulates GLP-1 secretion from enteroendocrine cells, enhances glucose uptake in 3T3-L1 adipocytes	[14]
AZ13581837	FFAR4	Increases insulin secretion and reduces blood glucose levels in mice	[15]
CpdA	FFAR4	Increases insulin sensitivity and improves glucose tolerance in mice	[16]
Metabolex-36	FFAR4	Insulin secretagogue with glucose-lowering properties	[17]
GSK137647	FFAR4	Improves glucose tolerance	[18]
TUG-1197	FFAR4	Enhances insulin sensitivity and reducesbody weightt	[19]
NCG21	FFAR4	Increases GLP-1 secretion	[20]
GW9508	FFAR4	Enhances insulin sensitivity and thermogenic activity of adipocytes	[21]

## Data Availability

Not applicable.

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
