# Peer review of "Free Fatty Acid Receptors (FFARs) in Adipose: Physiological Role and Therapeutic Outlook"

_cells, 2022, doi:10.3390/cells11040750_

Round 1

Reviewer 1 Report

Line 59 - subtitle suggests that only FFAR2 and FFAR4 will be described in the paragraph however lines 60-65 refers specifically to FFAR1. I would suggest to either change the title or move these several lines to a different part of the manuscript

lines 65-75 - I would suggest to combine these two sentences; you could start the sentence with the occurrence of FFAR2 in adipose tissue as this is the aim of this paragraph

lines 67-72 - similarly to my previous comment - please either rewrite the whole paragraph and focus on FFAR2 and 4 or rephrase the subtitle as in these lined major emphasis was put on FFAR 3 and 4

Fig 2 - the figure does not provide new informaiton to the reader; it summarises what has already been written in the text. The expression of FFAR2 is not only high in adipose tissue but also in
spleen, pancreas and peripheral blood mononuclear cells. In addition, from this figure the reader could presume that the expression of certain FFARs e.g. FFAR 1, 2 and 3, is equally abundant in enteroendocrine cells. I think the authors should rather show the level of abundancy of each FFARs for every organ so the figure provide new information 

Lines 86-91 - two contradicting sentences. First says that there was no significant effect on adipogenesis, whereas the second sentence indicates that there was no effect . Please clarify or rephrase both sentances

Lines 74-97 - I would suggest to rearrange the observations/summaries from studies so that  first you describe results from humans and then animals (murine) 3T3 L1.

Line 100-102 - was the inhibitory effect the same for acetate and priopionate 

Lines 103-104 - 'in vivo infusion of acetate' - what route was the acetate infused. have the authors verified other way of administration and its effect on the lipolysis?

Lines 108-109 - could you provide more information about the synthetic agonists and other synthetic agonists- are there any similarities in the composition/mechanism of action?

Line 115- i would suggest to indicate that this is the fisr study that you are describing

Line 117 - please change FFAr2 to FFAR2

Lines 118-119 - the author include 'conversely' in the sentece whereas the sentence also describes the FFAR2 KO mice 

Line 120 - what does it mean 'normal' diet. 

Lines 112-126 - paragraph should state at the early begining that one of the studies used two different diets - regular and HF diet. Has the other study also compared different diets? what diet were the mice fed in the lines 114-117?

Paragraph line 130 - again it would be good to read first results from humans, then animals

Line 140- what are the 'specific agonists'

Line 143- I would suggest to change the explanation of the abbreviation BAT to brown adipose tissue

Line 150 - explain abbreviation FGF21 as it appears for the first time

Line 151 - the author uses KO- previous in the text the authors used knockout. Pleas align

Line 160 - should be 'clinical'. If possible please rephrase the sentence as its difficult to read

Fig 3- not very informative (again description of what already been mentioned in the text)

Lines 222-227 - duplication of informaiton from the Abstract and introduction. suggest to delete it 

conclusion to be rewritten as it doesnt provide any information or thoughts on further research. 

Author Response

Thank you for a comprehensive evaluation of our manuscript. Here is a point by point response to your queries. The changes are highlighted in the manuscript (blue)

Review 1.  Line 59 - subtitle suggests that only FFAR2 and FFAR4 will bepoint-by-point described in the paragraph however lines 60-65 refers specifically to FFAR1. I would suggest to either change the title or move these several lines to a different part of the manuscript

  • This section has been modified to focus on FFAR2 and FFAR4

ines 65-75 - I would suggest to combine these two sentences; you could start the sentence with the occurrence of FFAR2 in adipose tissue as this is the aim of this paragraph

  • Done

ines 67-72 - similarly to my previous comment - please either rewrite the whole paragraph and focus on FFAR2 and 4 or rephrase the subtitle as in these lined major emphasis was put on FFAR 3 and 4

  • This section has been modified to focus on FFAR2 and FFAR4

Fig 2 - the figure does not provide new informaiton to the reader; it summarises what has already been written in the text. The expression of FFAR2 is not only high in adipose tissue but also in
spleen, pancreas and peripheral blood mononuclear cells. In addition, from this figure the reader could presume that the expression of certain FFARs e.g. FFAR 1, 2 and 3, is equally abundant in enteroendocrine cells. I think the authors should rather show the level of abundancy of each FFARs for every organ so the figure provide new information 

  • The Figure has been deleted and instead a table is included to show the tissues where each FFAR isoform is enriched

Lines 86-91 - two contradicting sentences. First says that there was no significant effect on adipogenesis, whereas the second sentence indicates that there was no effect . Please clarify or rephrase both sentences

  • Both sentences demonstrate that FFAR2 is not involved in adipogenesis. For clarity, the word “significant” has been deleted.

Lines 74-97 - I would suggest to rearrange the observations/summaries from studies so that  first you describe results from humans and then animals (murine) 3T3 L1.

  • This section starts with describing the expression in human adipose followed by mouse adipose and 3t3 L1 adipocytes. The impact of FFAR2 on adipogenesis is arranged in such a way so as to first describe those studies that show positive effects.

Line 100-102 - was the inhibitory effect the same for acetate and priopionate 

  • Yes, acetate and propionate showed similar inhibitory effects on lipolysis

Lines 103-104 - 'in vivo infusion of acetate' - what route was the acetate infused. have the authors verified other way of administration and its effect on the lipolysis?

  •   Sodium acetate was injected via the intraperitoneal (IP) route into mice. The authors did not mention any other study using other ways of administration of acetate and its effect on lipolysis.

Lines 108-109 - could you provide more information about the synthetic agonists and other synthetic agonists- are there any similarities in the composition/mechanism of action?

  • The synthetic agonist used in this study is a phenylacetamide derivative, which the authors name as compound 58. Other synthetic agonists are discussed in detail in the next section.

Line 115- i would suggest to indicate that this is the fisr study that you are describing

  • Done

Line 117 - please change FFAr2 to FFAR2

  • Done

Lines 118-119 - the author include 'conversely' in the sentence whereas the sentence also describes the FFAR2 KO mice 

  • The sentence describes Ffar2 KO mice and the previous sentence uses mice overexpressing Ffar2.

Line 120 - what does it mean 'normal' diet. 

  •                  A normal diet is a regular diet fed to laboratory mice in which, about 60% of calories come from carbohydrates, 20% each from proteins and fat. In a high-Fat diet, 60% of the calories come from fat and 20% each from proteins and carbohydrates

Lines 112-126 - paragraph should state at the early beginning that one of the studies used two different diets - regular and HF diet. Has the other study also compared different diets? what diet were the mice fed in the lines 114-117?

  • The first study used only a normal diet whereas the second study used both normal and high-fat diets. The sentences are rephrased to bring clarity.

Paragraph line 130 - again it would be good to read first results from humans, then animals

  • Done

Line 140- what are the 'specific agonists'

  • TUG-891 and GW9508

Line 143- I would suggest to change the explanation of the abbreviation BAT to brown adipose tissue

  • Done

Line 150 - explain abbreviation FGF21 as it appears for the first time

  • Done

Line 151 - the author uses KO- previous in the text the authors used knockout. Pleas align

  • Done

Line 160 - should be 'clinical'. If possible please rephrase the sentence as its difficult to read

  • Done

Fig 3- not very informative (again description of what already been mentioned in the text)

  • The figure just summarized the role of FFAR in adipose metabolism

Lines 222-227 - duplication of informaiton from the Abstract and introduction. suggest to delete it 

conclusion to be rewritten as it doesnt provide any information or thoughts on further research. 

  • The conclusion has been modified to avoid duplication and to provide views on further research.

Reviewer 2 Report

Major:

  1. The English needs editing throughout. It is understandable, but there are numerous errors.
  2. The review is rather limited in scope. For example, there is no discussion of the signaling pathways or mechanisms involved in any of the responses to FFAR agonists.
  3. Figure 2 seems overly simplistic. There are no references provided for the "yes" and "no" labels, and it is not clear whether only human tissues are included here. There is no figure legend to explain the content. All of the FFARs are fairly widely expressed.
  4. Figure 3 needs some work. The human and mouse parts of the figure refer to variants or knockouts, and show the effects of these changes. The part about adipocytes just says "adipocytes", but they has an arrow indicating that some change occurred...what caused the change?
  5. Page 5, lines 176-177: Did the drug actually cause liver toxicity, or just change biomarkers in a way that suggested possible toxicity?
  6. Page 5, line 180: should be "has", not "have"
  7. Table 1: For TUG-891, it is not clear why effects on inflammatory mediators are mentioned here. The drug has many effects on various types of cells, and to be consistent only effects related to metabolism should be listed here.

Minor:

  1. Page 1, lines 42-43: The sentence beginning "Free fatty acids..." does not add anything that was not already said, and should be removed.
  2. Page 3, lines 84-85: The sentence doesn't make sense as written. Should "unregulated" be "dysregulated"?
  3. Page 5, line 159:  should be "act", not "acts".

Author Response

Thank you for a comprehensive evaluation of our manuscript. Here is a point-by-point response to your queries. The changes are highlighted in the manuscript (red)

  1. The English needs editing throughout. It is understandable, but there are numerous errors.
  • We have edited the manuscript and corrected the grammatical mistakes
  1. The review is rather limited in scope. For example, there is no discussion of the signaling pathways or mechanisms involved in any of the responses to FFAR agonists.
  • The signaling pathways activated by FFAR ligands have been added to the manuscript
  1. Figure 2 seems overly simplistic. There are no references provided for the "yes" and "no" labels, and it is not clear whether only human tissues are included here. There is no figure legend to explain the content. All of the FFARs are fairly widely expressed.
  • We have deleted the figure and instead included a table listing human tissues where each FFAR isoform is enriched along with the references
  1. Figure 3 needs some work. The human and mouse parts of the figure refer to variants or knockouts, and show the effects of these changes. The part about adipocytes just says "adipocytes", but they has an arrow indicating that some change occurred...what caused the change?
  • Human part refers to the variant that is associated with increased risk of obesity and diabetes and the mouse part of the figure refers to the knockout. In vitro results are from both human and murine adipocytes.
  1. Page 5, lines 176-177: Did the drug actually cause liver toxicity, or just change biomarkers in a way that suggested possible toxicity?
  • Yes TAK-875 did cause liver toxicity in human subjects. TAK-875-induced liver injury is believed to be independent of FFAR1 and may be mediated through formation of hepatic acyl glucuronide metabolites

  1. Page 5, line 180: should be "has", not "have"
  • Corrected
  1. Table 1: For TUG-891, it is not clear why effects on inflammatory mediators are mentioned here. The drug has many effects on various types of cells, and to be consistent only effects related to metabolism should be listed here.
  • Corrected
  1.  

Minor:

  1. Page 1, lines 42-43: The sentence beginning "Free fatty acids..." does not add anything that was not already said, and should be removed.
  • Corrected

  1. Page 3, lines 84-85: The sentence doesn't make sense as written. Should "unregulated" be "dysregulated"?
  • It was a typo. The word has been corrected to upregulated
  1. Page 5, line 159: should be "act", not "acts".
  • Corrected

Reviewer 3 Report

This is a well-written review which covers the overall data of FFARs. For minor points, I think it would be better to present the clinical trial stages in Table 1 as well as the primary outcomes of trials. 

Author Response

Reviewer 3

This is a well-written review which covers the overall data of FFARs. For minor points, I think it would be better to present the clinical trial stages in Table 1 as well as the primary outcomes of trials. 

  • Thank you for the suggestion but none of the FFAR2 or FFAR4 agonists have yet reached clinical trials

Round 2

Reviewer 1 Report

Thank you for addressing my previous comments. I identified several other minor aspects which should be addressed to improve the quality of the manuscript

line 46 and 73 - duplicated information "palmitate, oleate, and linoleate'. please consider deleting one of them

Line 149 - I would suggest to change the 'normal diet' to either a 'standard chow diet" or "regular chow" 

Line 183-184 - BAT - stands for brown adipose tissue. Please update in the text - this change hasnt been implemented to the revised version although I provided this comment before

Line 168- a deleterious variant of FFAR4 is described as R270H; however, fig 2 shows RH270. Please update  

Line 168 - please replace 'healthy ones' with 'healthy individuals' or 'normoglycemic controls'

Line 192-  sometimes the authors uses KO sometimes knockout. Please align. If the authors chooses to use KO- the abbreviation should be explained in the text and used consistently throughout the manuscript

Reference to Figure 1 is missing in the text. The reference should be added in the following paragraph 'he downstream signal mediated by FFAR2 and FFAR4 in adipose tissue' (lines 199-215). 

 lines 199-215- the following paragraph entitled 'The downstream signal mediated by FFAR2 and FFAR4 in adipose tissue' should be placed before section ''3. Role of FFAR2 in adipose metabolism and energy homeostasis"

line 291 - Please change section 'Conclusion' from '6' to '7'. Section 6 refers to 'FFAR agonists in the treatment of metabolic diseases' 

line 296 - please remove multiple spaces 

lines 297-299 - please rewrite the sentence. The authors could emphasize that FFARs are regarded as targets for novel drugs rather than using the following phrase 'FFAR based drugs' (this phrase should be avoided).

lines 297-302 please rewrite  and consider combining into one comprehensive sentence  

Author Response

Thanks for a comprehensive review of our manuscript.

Here is the point-by-point response to the queries raised. 

line 46 and 73 - duplicated information "palmitate, oleate, and linoleate'. please consider deleting one of them

  • Done

Line 149 - I would suggest to change the 'normal diet' to either a 'standard chow diet" or "regular chow"

  • Done

Line 183-184 - BAT - stands for brown adipose tissue. Please update in the text - this change hasnt been implemented to the revised version although I provided this comment before

  • Done

Line 168- a deleterious variant of FFAR4 is described as R270H; however, fig 2 shows RH270. Please update 

  • Done

Line 168 - please replace 'healthy ones' with 'healthy individuals' or 'normoglycemic controls'

  • Done

Line 192-  sometimes the authors uses KO sometimes knockout. Please align. If the authors chooses to use KO- the abbreviation should be explained in the text and used consistently throughout the manuscript

  1. Done. The text has been modified and “knockout” used throughout

Reference to Figure 1 is missing in the text. The reference should be added in the following paragraph 'he downstream signal mediated by FFAR2 and FFAR4 in adipose tissue' (lines 199-215).

  • Done

 lines 199-215- the following paragraph entitled 'The downstream signal mediated by FFAR2 and FFAR4 in adipose tissue' should be placed before section ''3. Role of FFAR2 in adipose metabolism and energy homeostasis"

  • Section 3 and section 4 describe the role of FFAR2 and FFAR4 in adipose metabolism followed by section 5 to describe the downstream signaling cascades. I believe this section is better placed after section 4.

line 291 - Please change section 'Conclusion' from '6' to '7'. Section 6 refers to 'FFAR agonists in the treatment of metabolic diseases

  • Done

line 296 - please remove multiple spaces

  • Done

lines 297-299 - please rewrite the sentence. The authors could emphasize that FFARs are regarded as targets for novel drugs rather than using the following phrase 'FFAR based drugs' (this phrase should be avoided).

  • This section has been re-written to address the concerns

lines 297-302 please rewrite  and consider combining into one comprehensive sentence 

  • This section has been re-written to address the concerns

Reviewer 2 Report

Major:

  1. Figure 2 remains confusing. Perhaps if you inserted "FFAR4 wt" before "adipocytes" it would make sense.
  2. Page 6: The authors have added a section about G-protein-mediated signaling. This section is a good addition, although it is still rather simplistic. FFARs also signal via arrestins; this is not mentioned at all and the reviewer is not sure whether this plays a role in the metabolic signaling.

Minor:

1. Line 13:  suggest "In addition", instead of "Besides".

2. Lines 46 and 73: The names of the fatty acids should not be capitalized.

3. Line 50:  suggest "such as" instead of "like".

4. Line 54: should be "as well as in human trials."

5. Line 59: delete "free".

6. Line 64: should be "G-proteins"; remove "complex". Do not capitalize "protein" here.

5. Line 66:  should be "Approximately" instead of "Around".

6. Line 67: "Vision" should not be capitalized.

7. Line 73: should be "such as" (or "including") and not "like"

8. Legend to Figure 1: suggest using "selectively" and not "specifically". Few ligands are specific for a certain receptor.

9. Line 104: insert "the" before "cell line".

10. Line 106: use "In addition" rather than "Besides".

11. Line 110: remove the comma after "acetate".

12. Lines 111, 120, 129, 149, 236, 243:  add a period after "al". To avoid this issue, it is suggested that you use "and colleagues" or "and co-workers" instead of "et al." in a sentence.

13. Line 138: use the abbreviation "FFA"; the capitalization is incorrect in the current wording anyway.

14. Line 147:  probably "ffar2" should be italicized, if it is the gene name.

15. Line 149: delete "on".

13. Line 119:  use "did not" rather than "didn't".

13. Line 122: should be "and that this effect was..."

14. Line 128: delete "the".

15. Line 159: insert "The" before "R270H".

16. Line 170: should be "act" and not "acts".

17. Line 174: delete "develop" before "insulin resistance".

18. Lines 177-178: do not capitalized "glucose" and "insulin".

19. Line 191: delete the apostrophe after "neonates"; perhaps this should be "neonatal" instead.

20. Line 193: delete "the".

21. Line 195: should be "summarizes" and not "summarized".

22. Line 213: which "FFAR"?

23. Line 217: should be "Ever" and not "Even".

24. Line 227: delete "R".

25. Line 236: delete the comma after "Lee".

26. Line 239: delete "activation of".

27. Line 240: add a comma after "CMTB"; also should be "...distinct from that of the..."

28. LINE 241: "R" should be capitalized in "FFAR".

29. Line 246: there should be a space between "compound" and "1".

30. Line 248: "release" should not be capitalized.

31. Table 2, first line: "TAK-875" should not be capitalized.

32. Line 295:  remove comma after "while". Suggest re-wording to "While much work has been done..."

33. Line 298: gene names should be italicized.

34. In the graphical abstract, suggest removing the question mark after "therapeutic drugs". It makes the entire abstract look questionable. Perhaps change to "...as potential therapeutic drugs" with no question mark.

Author Response

Thanks for the revision. we have addressed the concerns of the reviewer and made the necessary changes in the manuscript (highlighted in red)

Reviewer 3 Report

I think the current manuscript is sufficient to be accepted.

Author Response

Thank you

Round 3

Reviewer 2 Report

Thank you for the edits. Figure 2 is still not making sense to me as presented. It shows "adipocytes" and then below says that various responses were increased (as shown by an arrow). In the other cases, there was a manipulation (such as a mutation or knockdown), and then the arrows indicate what happened after the manipulation. For the adipocytes, there is again an arrow indicating that responses were increased, but it is not clear that you are talking about what happens when wt FFAR4 is activated. Please add something else to explain...perhaps "FFAR4 activation in adipocytes" would work.

Author Response

Thanks for reviewing our paper. Figure 2 has been modified as per your suggestion.